# Sensitiveness of Variables Extracted from a Fitness Smartwatch to Detect Changes in Vertical Impact Loading during Outdoors Running

**DOI:** 10.3390/s23062928

**Published:** 2023-03-08

**Authors:** Cristina-Ioana Pirscoveanu, Anderson Souza Oliveira

**Affiliations:** 1Department of Health Science and Technology, Aalborg University, 9210 Aalborg, Denmark; 2Department of Materials and Production, Aalborg University, 9220 Aalborg, Denmark

**Keywords:** running-related injuries, outdoors running, tibial acceleration, wearables, shock absorption

## Abstract

Accelerometry is becoming a popular method to access human movement in outdoor conditions. Running smartwatches may acquire chest accelerometry through a chest strap, but little is known about whether the data from these chest straps can provide indirect access to changes in vertical impact properties that define rearfoot or forefoot strike. This study assessed whether the data from a fitness smartwatch and chest strap containing a tri-axial accelerometer (FS) is sensible to detect changes in running style. Twenty-eight participants performed 95 m running bouts at ~3 m/s in two conditions: normal running and running while actively reducing impact sounds (silent running). The FS acquired running cadence, ground contact time (GCT), stride length, trunk vertical oscillation (TVO), and heart rate. Moreover, a tri-axial accelerometer attached to the right shank provided peak vertical tibia acceleration (PK_ACC_). The running parameters extracted from the FS and PK_ACC_ variables were compared between normal and silent running. Moreover, the association between PK_ACC_ and smartwatch running parameters was accessed using Pearson correlations. There was a 13 ± 19% reduction in PK_ACC_ (*p* < 0.005), and a 5 ± 10% increase in TVO from normal to silent running (*p* < 0.01). Moreover, there were slight reductions (~2 ± 2%) in cadence and GCT when silently running (*p* < 0.05). However, there were no significant associations between PK_ACC_ and the variables extracted from the FS (r < 0.1, *p* > 0.05). Therefore, our results suggest that biomechanical variables extracted from FS have limited sensitivity to detect changes in running technique. Moreover, the biomechanical variables from the FS cannot be associated with lower limb vertical loading.

## 1. Introduction

Accelerometry has become a popular method to assess human movement in natural conditions, being used in investigations dedicated to improving the quality of life and performance in various populations [1,2]. However, running biomechanical variables extracted from accelerometers might differ from those extracted from gold-standard laboratory equipment [3]. Tibial acceleration is a good surrogate for running impact properties, such as vertical ground reaction forces [3,4], with the advantage of allowing data acquisition in ecological running conditions [5]. Improvements in accelerometry data accuracy are still necessary, as non-standardized sensor location and placement might introduce undesired data variability [6,7]. Studies using accelerometry have revealed reduced vertical tibial acceleration during indoor treadmill/overground running when compared to outdoor conditions, such as grass or asphalt [3], and has been used as an alternative assessment tool of vertical loading rates [4]. Furthermore, peak tibial acceleration is a useful parameter for the assessment of running-related injuries [8,9]. Therefore, tibial acceleration is a highly relevant method to assess a runner’s impact properties in natural conditions. 

The use of accelerometers has been popularized through commercial products such as wearable tracking monitors [5,10,11]. In-built accelerometers in smartwatches and/or external chest straps allow measuring running biomechanical parameters [12,13]. The popularity of the technology has grown, since more than 66% of runners use wearable devices to quantify their running performance, such as Garmin^®^ watches that are used by ~44% of runners [11,14,15]. Interestingly, whereas most runners track basic training parameters such as running time, speed, and heart rate, less than 9% track biomechanical parameters such as cadence [16]. One reason for the low adherence to using such metrics is the lack of understanding of their meaning [17,18,19]. Assessing running biomechanics outdoors through wearable devices has been shown to be valid and reliable, although small variations in reliability could be present between different devices [20]. Therefore, evaluating whether the running biomechanical parameters provided by commercial smartwatches are capable of tracking changes in running technique may help runners to adopt such metrics to evaluate their training progression. 

An example of using biomechanical data to assess performance can be found in gait re-training, which is a relevant method to prevent running-related injuries through modifications in running technique [21,22]. It has been widely shown that vertical impact loading is associated with some types of running-related injuries, such as stress fractures [21,23,24], patellofemoral pain, and plantar fasciitis [25]. Running coaches can teach novice runners to minimize their impact forces by listening to the sounds runners produce [26]. Running with lower footfall sounds reduces peak vertical forces and loading rates, as well as changing the runner’s foot strike technique [27,28,29]. Achieving ideal reductions in vertical impact loading while naturally performing a running workout may be facilitated through objective measurements such as accelerometry. Therefore, it is highly relevant to establish whether current smartwatches that offer assessment of running biomechanics are sensitive to detect changes in vertical impact loading. Such sensitiveness can be attested by comparing results from traditional vertical tibial acceleration to the results extracted from a commercially available smartwatch, which uses chest accelerometry to derive running biomechanical parameters.

In this study, a commercial smartwatch paired with a chest strap and a tibial-worn accelerometer were used during normal and silent running bouts. The purpose was to investigate whether the changes in running technique (normal vs. silent running) could be detected by variables extracted from the smartwatch and chest strap combination (further referred to as a wearable sensor [WS]). Moreover, we investigated whether there was an association between tibial acceleration and the variables from the WS. Our main hypothesis was that the running biomechanical data provided by the WS are sensitive to detect changes in running technique and can also be associated with tibial acceleration.

## 2. Methods

### 2.1. Participants

Twenty-eight recreational runners (21 male, 7 female), right foot dominant (age: 25 ± 2.5 years, height: 181.7 ± 9.1 cm, weight 77.1 ± 12.7 kg, 10 ± 4 years running experience, 21 ± 13 km weekly running volume) agreed to participate in the study. Initial contact technique based on heel contact (e.g., rearfoot running) was part of our inclusion criterion. Participants were injury-free for a minimum of 6 months before the test, as well as avoided performing any strenuous exercise 24 h before participating in this study. Additionally, participants were asked to avoid consuming any product containing caffeine and alcohol for at least 12 h before the test. Participants were informed about the experimental procedure and provided verbal and written informed consent to participate in this study. The procedures applied in this study were approved by the local ethical committee (Region Nordjylland, Denmark). All methods were carried out in accordance with relevant guidelines and regulations from the Declaration of Helsinki (2004).

### 2.2. Experimental Setup

In a single session, participants were asked to run using two different running styles (normal and silent) on an outdoor 400-m running track using their preferred regular running shoes. Previous studies have shown that silent running is predominantly achieved by changing running technique towards forefoot strike [26,29], allowing the comparison between different types of running techniques in this study. For both conditions, runners were asked to perform 2 × 95-m running bouts at 3 m/s. This running speed was chosen as it is a commonly used speed in running biomechanical studies [30]. The running speed was continuously measured through the embedded GPS on the smartwatch. Moreover, the consistency of the running speed throughout the test was assured by the experimenters checking if the runner was within ±2 s of the expected time to complete the 95 m bouts. If the running speed was not within ±2 s, the trial was excluded, and another running bout was performed. The running bouts started 10 m prior to the straight lines on the running track, allowing runners to achieve a constant speed throughout the ~85 m straights. After running 95 m, runners were asked to walk during the turn on the track until reaching the mark to start running again (see Figure 1A for illustration). The sequence for executing running conditions (normal or silent) was randomized for each runner. Silent running was defined as running with the lowest impact sound by minimizing landing noises. Participants were instructed to run as they normally do, but with the lowest impact sound possible at the pre-established speed [26,27,28]. Therefore, runners were not instructed on how to change their running technique to reduce impact sounds. 

Prior to data recording, all participants were provided with a warm-up session consisting of run-throughs, walking lounges, running with high knees, and leg swings [26,29]. Subsequently, a minimum 10-m familiarization period was specifically conducted for silent running at the running track. Participants were considered familiarized when they could consistently generate low-impact sounds, qualitatively assessed by the experimenter, at the target running speed [29].

### 2.3. Data Acquisition and Analysis

The fitness smartwatch (Garmin Forerunner 735XT, Garmin International, Kansas City, MO, USA) was located on the right wrist and provided the running speed through the embedded GPS. A compatible chest strap containing a tri-axial accelerometer acquired the heart rate and cadence, vertical oscillation, stride length, and ground contact time through the embedded accelerometer. All WS data were sampled at 1 Hz. The data from this type of device had been previously validated for the assessment of heart rate [13,31], as well as running cadence, ground contact time, and vertical oscillation [32,33].

All data processing in this study was performed using custom-made scripts (Matlab 2020b, The Mathworks Inc., Natick, MA, USA). The data were imported into Matlab and the sectors representing running were manually selected, by identifying the periods of nearly constant running speed at 3 m/s. The heart rate of each runner was normalized by their predicted maximum heart rate, following the equation HR = 208 − (0.7 × Age) [34]. The heart rate and spatio-temporal running parameters from each runner were represented by averaging all data from such variables across the two bouts of running. In addition, we defined the variability in the heart rate and spatio-temporal running parameters variables using the coefficient of variation (CV, defined as the ratio of the standard deviation to the mean). In Figure 2, data from a representative runner is shown for running speed (Figure 2A), trunk vertical oscillation (Figure 2B), and heart rate (Figure 2C) during a full lap on the 400-m track consisting of two bouts of running and two bouts of walking periods. The shaded gray areas indicate the two bouts of normal running (blue line) and silent running (red line). In Figure 2D mean (solid line) and ±1 standard deviation (shaded area) vertical right tibial acceleration during normal (92 steps) and silent running (93 steps).

Vertical tibial acceleration was captured from the right limb using a wearable tri-axial inertial measurement unit sensor (Shimmer3, ShimmerSensing, Dublin, Ireland), sampled at 1024 Hz. The sensor was strapped to the antero-medial side of the right tibia at 1/3 length from the knee joint. The accelerometer was oriented in such a way that the vertical axis of the accelerometer coincided with the longitudinal axis of the tibia (Figure 1B). Data was stored on a memory card in the accelerometer and downloaded offline for further processing. 

The vertical shank accelerometer data were firstly lowpass filtered (60 Hz) and converted from linear acceleration (m/s^2^) to *g* force. The data sectors representing running were manually selected, by identifying the periods of nearly constant peak acceleration. Subsequently, the peak acceleration events related to foot contact with the floor were defined during the running bouts at constant running speed. For each runner, a total of 90 ± 8 and 91 ± 14 peak vertical acceleration (PK_ACC_) events were included for normal and silent running, respectively. In addition, we calculated the intra-subject CV across the single-trial PK_ACC_ data. Figure 2D illustrated a typical comparison between PK_ACC_ from normal and silent running.

The rate of perceived exertion (arbitrary unit [a.u.]) was assessed using the Borg rate of perceived exertion (RPE) scale (intensity starting at 6 ‘easy’ to 20 ‘maximal exertion’) after every 95 m running bout [35]. A 100 cm × 60 cm scale was fixed at a 1.80 m height beside the track in the collecting points. Participants were instructed to say the number from the Borg scale that represented their current effort level.

### 2.4. Statistical Analysis

All statistical analyses were performed using custom-made scripts (Matlab 2020b, The Mathworks Inc., Natick, MA, USA). The normality of the dependent variables was assessed using Kolmogorov-Smirnov tests. Differences between normal and silent running on the dependent variables (e.g., running speed, vertical tibial acceleration, trunk vertical oscillation, cadence, ground contact time, stride length, heart rate, and rate of perceived exertion) were assessed using two-tailed, paired t-Student tests. Cohen’s D effect size was computed for all comparisons, where a value of 0.2 was considered small, 0.5—medium, and 0.8—large [36]. Pearson correlations were calculated using PK_ACC_ across all runners between normal versus silent running. In addition, Pearson correlations were calculated between PK_ACC_ and trunk vertical oscillation, cadence, ground contact time, stride length, and heart rate for both normal and silent running. Finally, the delta between normal and silent running (as % difference) was calculated from PK_ACC_ and the running biomechanical variables (trunk vertical oscillation, cadence, ground contact time, stride length). Subsequently, Pearson correlations were calculated between the delta PK_ACC_ and the delta from the biomechanical variables. An alpha level of significance *p* < 0.05 was selected for all statistical tests.

## 3. Results

No differences in running speed were found between normal (3.0 ± 0.1 m/s) and silent running (3.0 ± 0.2 m/s). In addition, the RPE was greater during silent running (9.7 ± 1.7 a.u.) when compared to normal running (8.4 ± 1.1 a.u.; *p* < 0.01, effect size = 0.38).

### 3.1. Tibial Acceleration and Spatio-Temporal Running Parameters

Running silently reduced the PK_ACC_ from 8.1 ± 2.3 g to 7.1 ± 2.9 g (13 ± 19% reduction, *p* < 0.005, effect size = 0.40, Figure 3A). Moreover, vertical oscillation was increased from normal running (0.102 ± 0.017 m) to silent running (0.107 ± 0.07 cm, 5 ± 10% increase, *p* < 0.01, effect size = 0.30, Figure 3B). Regarding other running mechanical parameters, silent running was achieved with reduced cadence (1.3 ± 3.1% reduction, *p* < 0.05, effect size = 0.22, Figure 3C) and reduced ground contact time (1.8 ± 4.7% reduction, *p* < 0.05, effect size = 0.23, Figure 3D). However, no changes in stride length were found between conditions (*p* > 0.05, effect size = 0.01, Figure 3E). The heart rate during silent running was higher (71.7 ± 8.4% max) when compared to normal running (69.5 ± 8.5% max, 3.3 ± 5.6% increase, *p* < 0.01, effect size = 0.25, Figure 3F). 

The PK_ACC_ presented the expected reduction during silent running for 23 out of 28 runners (~82% of runners). In contrast, the trunk vertical oscillation, cadence, and ground contact time presented inconsistent inter-subjects data patterns. Trunk vertical oscillation was increased for 65% of runners, whereas running cadence and ground contact time were reduced for 56% and 65% of runners, respectively.

### 3.2. Inter-Subject Variability 

The intra-subject variability analysis demonstrated a greater CV during silent running when compared to normal running for PK_ACC_ (*p* < 0.001, Table 1) and running speed (*p* < 0.05). In addition, the CV for heart rate was reduced during silent running when compared to normal running (*p* < 0.05). No differences in CV were found for cadence, trunk vertical oscillation, stride length, and ground contact time.

### 3.3. Association Tibial Acceleration vs. Spatio-Temporal Running Parameters

The PK_ACC_ from normal and silent running were highly correlated (Figure 4A). However, there were no significant associations between PK_ACC_ and the variables extracted from the WS (*p* > 0.05, Figure 4B–F).

The delta change in PK_ACC_ between normal and silent running was poorly correlated to trunk oscillation (r = 0.003, *p* > 0.05, Figure 5A), cadence (r = 0.02, *p* > 0.05, Figure 5B), ground contact time (r = 0.23, *p* > 0.05, Figure 5C), and stride length (r = 0.23, *p* > 0.05, Figure 5D).

## 4. Discussion

The main purpose of this study was to determine whether running biomechanical data acquired through a popular commercially available fitness smartwatch (Garmin Forerunner 735XT^®,^ Olathe, KS, USA) and compatible chest strap are sensitive to detecting within-subject changes in running technique, potentially corroborating results provided by shank accelerometry. Silent running is predominantly achieved by modifying running technique from rearfoot to forefoot strike [26,29], a change previously correlated with reduced peak tibial shocks [4]. Therefore, our main assumption was that tibial acceleration would provide relevant benchmarking regarding reductions in vertical impact loading between normal and silent running, and the spatio-temporal variables from the smartwatch and chest strap were compared to the benchmark measurement.

Our main results showed that vertical tibial acceleration was reduced by ~13% across all runners when attempting to reduce footfall sound volume. However, the significant differences between normal and silent running in other variables only reached a marginal 2–5% difference. Therefore, variables such as cadence and ground contact time present statistical differences, but are not meaningful for determining differences in running technique. Moreover, none of the variables extracted from the smartwatch and chest strap demonstrated an acceptable association to the shank acceleration data (see Figure 4), both in isolated conditions (normal or silent running) and when expressed as delta changes from normal to silent running (see Figure 5). In addition, runners could reduce their vertical tibial acceleration by 13% during silent running at a similar speed, but heart rate increased from ~69% to ~72% maximum heart rate, and the Borg scale increased by ~1.4 points. This result demonstrated the increased effort to modify running style at a fixed running speed. Therefore, our results suggested detectable physiological changes and perceived effort parameters when modifying running style from normal to silent running. However, running biomechanical parameters extracted from the smartwatch and chest strap does not offer information that can be associated with shank vertical impact loading.

Lower leg accelerometry can provide a more accurate representation of the vertical impact loading sustained by distal lower limb segments, as the measurement is closer to the bone strain and stresses sustained through repeated ground impacts [35]. Conversely, chest accelerometry acquired with smartwatches can be influenced by the kinetic chain (lower legs, pelvis, and spine), which can provide shock attenuation and, as such, influence the level of oscillations captured [37]. Moreover, the data extracted from the chest-mounted accelerometer in this study were processed by the smartwatch software, and the data output for the relevant biomechanical variables was expressed at 1 Hz. It is noteworthy that there was a discrepancy in data processing between tibial accelerometry and the biomechanical data extracted from the smartwatch and chest strap, which is a strong limitation in determining the true quality of the smartwatch and chest strap accelerometer data. 

Although chest and wrist-worn accelerometers are reliable and present low sensitivity to sensor error positioning [38], tibial acceleration may provide the ideal method to assess impact loading and its magnitude [37]. Our study corroborates such suggestions, as the tibial acceleration was the most sensitive variable to differentiate between the two investigated running techniques, and there was no association between tibial acceleration and the variables extracted from the smartwatch and chest strap. In addition, the observed statistical differences in biomechanical variables extracted from the smartwatch and chest strap were only marginal, and might not provide meaningful information to runners. Altogether, our results revealed that silent running is more demanding, but it does not induce dramatic changes within short bouts. Therefore, it may be possible to add short bouts of silent running into running training routines as a form of improving running performance. Further studies are necessary to demonstrate whether prolonged periods of silent running may cause more relevant changes in the investigated variables.

Runners can naturally achieve self-optimization during their training sessions to run longer and faster [5]. However, silent running bouts seem to disrupt the natural variability of their running gait, as demonstrated by an increased CV for tibial acceleration and running speed. Interestingly, the CV for the tibial acceleration was almost twice as great as any other variable during both running conditions. The higher CV from tibial acceleration may suggest a greater sensitivity to running-related changes when compared to other variables. Interestingly, this greater variability may be a positive factor, as it may indicate the continuous adaptability of the lower segments to vertical impacts. The observed changes in CV may indicate that altering running techniques during training increases the difficulty of maintaining a stable running speed. Furthermore, Milner et al. [3] reported that tibial acceleration is susceptible to changes in running conditions from indoors to outdoors, making it plausible that tibial acceleration variability could be greater during silent running. Moreover, the CV for heart rate was significantly lower during silent running, suggesting that the effort required to silently run may be more consistent across the activity, stabilizing the heart rate. 

Running while minimizing footfall sound volume induces reductions in vertical impact forces [27,29], as runners adjust their leg stiffness to accommodate mechanical demands [39]. However, the impact forces generated during initial contact are dissipated, especially through the pelvis and the spine [36]. Therefore, runners use a combination of passive elements (ligament deformation, muscular oscillations, increased knee flexion, and some degree of foot pronation) as well as active elements, such as increased muscle activation, to dissipate the impact load that moves through the kinetic chain [36]. Thus, trunk stability during running is strictly related to the runner’s ability to control and adjust those elements [36]. Therefore, the higher trunk vertical oscillation during silent running may suggest that the runners actively work to perform against gravity, likely reducing their running economy [40]. This assumption is further supported by the increased heart rate and RPE during silent running in our study. Nevertheless, our tibial acceleration results indicated that runners may increase the stability of their lower bodies as they focus on their performance. This fact could limit their shock absorption abilities to the legs to achieve lower impact sounds, while disregarding the rest of the kinetic chain involved in running [41]. Therefore, mastering running while minimizing footfall sounds may require more training to stabilize trunk position towards attenuating trunk vertical oscillations. This is supported by the fact that trunk vertical oscillation is one of the five kinematic aspects linked to performance and running economy [40]. 

Despite the execution of only 2 × 95 m short bouts of exercise in our study, the continuous data acquisition throughout these bouts allowed the extraction of data from ~90 strides from each runner for the comparison between conditions. The recording of ~ 90 running strides is superior to most studies conducted in running biomechanics, and conforms to recent recommendations on running biomechanical analyses [30]. Running speed may also be a relevant limitation in this study, since runners were instructed to perform at their preferred speed. Therefore, extrapolating our results to running speeds greater or lower than those executed in our experiment must be done with caution. Another limitation is that running technique (rearfoot or forefoot strike) was not quantified in this study, limiting the accuracy in stating that silent running was achieved by changing the running technique for all participants. Furthermore, it has been shown that running surface influences accelerometer variability [3]. Our results were acquired from running on a running track made of synthetic rubber/polyurethane, limiting the extrapolation of our results to other indoors or outdoors conditions. Finally, the results related to running smartwatches and chest straps are limited to the models used in this experiment. However, it is noteworthy that the computation of running variables from the chest strap is universal for all the smartwatches of a certain manufacturer that support pairing with the chest strap.

## 5. Conclusions

Our results suggested that modifying the running technique from normal to silent running demanded increased heart rate and perceived effort. In addition, the peak vertical tibial acceleration was reduced to achieve silent running. However, the running biomechanical variables extracted from the smartwatch/trunk acceleration (trunk vertical oscillation, cadence and ground contact time) were marginally sensible to detect differences in running technique when considering the group analysis. More importantly, there is a lack of association between the biomechanical parameters extracted from the smartwatch and peak vertical tibial acceleration. Therefore, data from trunk accelerometry accessed through smartwatches may not be adequate to monitor changes in running techniques.

## Figures and Tables

**Figure 1 sensors-23-02928-f001:**
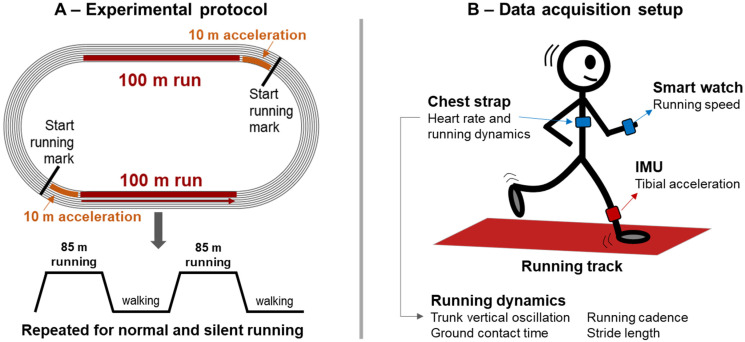
Illustration experimental setup used for the acquisition of overground running impact sounds (**A**), Overview of the data acquisition setting (**B**).

**Figure 2 sensors-23-02928-f002:**
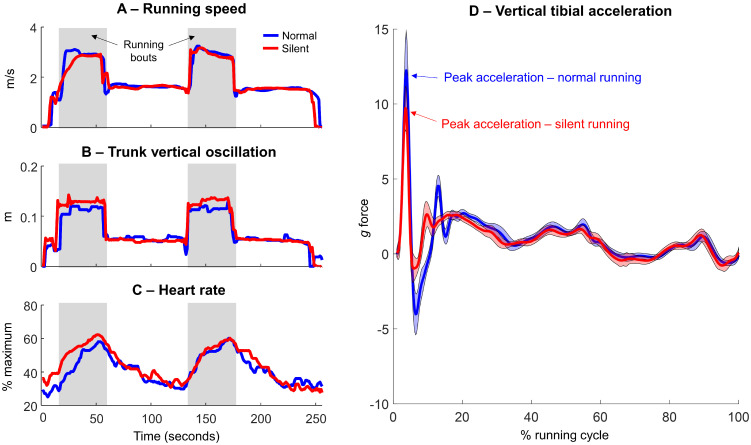
Single runner data illustrating running speed (**A**), trunk vertical oscillation (**B**), and heart rate (**C**). Shaded gray areas indicate the running bouts for both normal (blue) and silent running (red), mean (solid line) and ±1 standard deviation (shaded area) vertical right tibial acceleration (**D**).

**Figure 3 sensors-23-02928-f003:**
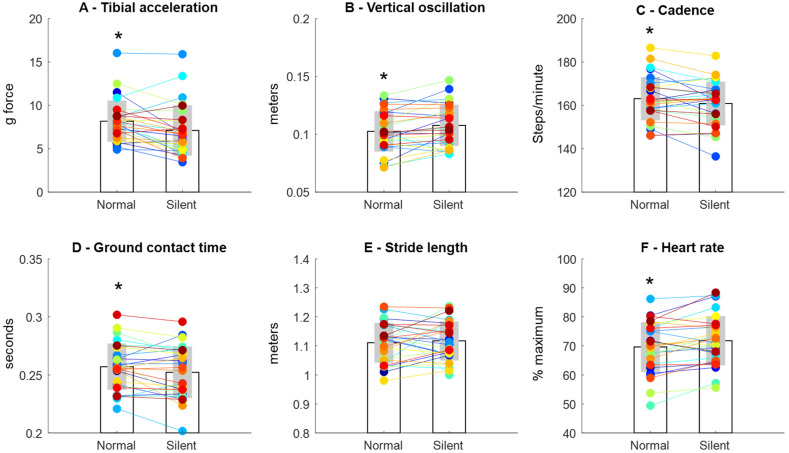
Mean (bar) and ±1 standard deviation (shade) tibial acceleration (**A**), vertical oscillation (**B**), cadence (**C**), ground contact time (**D**), stride length (**E**), and heart rate (**F**). Colored points and traces represent individual data and trend between conditions. * Denotes significant difference in relation to silent running (*p* < 0.05).

**Figure 4 sensors-23-02928-f004:**
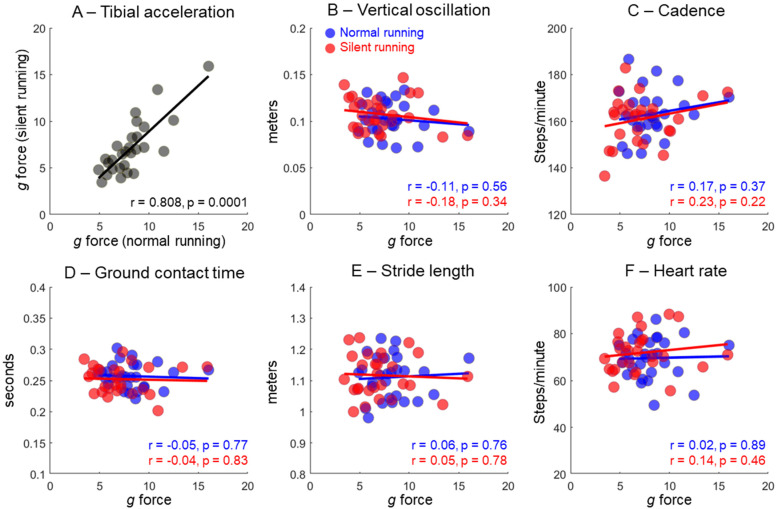
Pearson correlations between tibial acceleration in normal vs. silent running (**A**). Other panels illustrate Pearson correlations between peak vertical tibial acceleration and vertical oscillation (**B**), cadence (**C**), ground contact time (**D**), stride length (**E**), and heart rate (**F**) in normal (blue) and silent (red) running. Correlation coefficients (r) and significance levels (*p*) are described in the separate panels.

**Figure 5 sensors-23-02928-f005:**
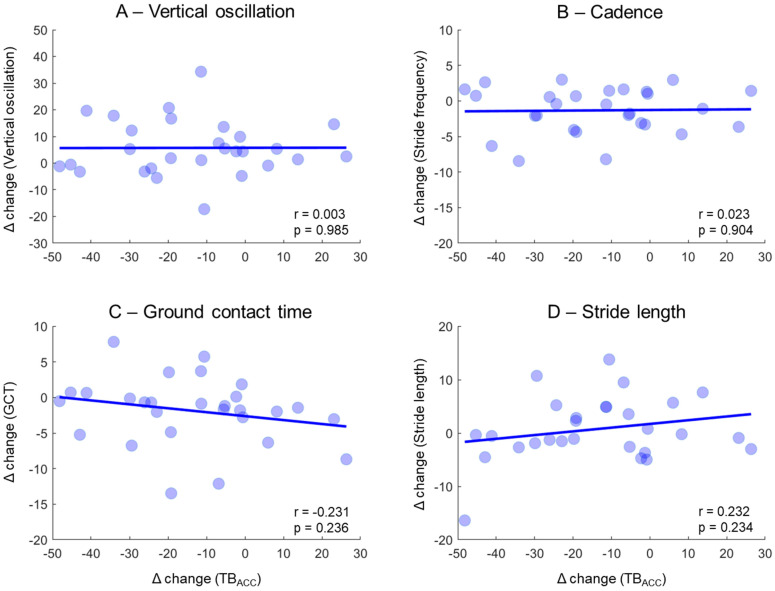
Pearson correlations between deltas from vertical tibial acceleration (TB_ACC_) and the deltas from trunk vertical oscillation (**A**), cadence (**B**), ground contact time (GCT) (**C**), and stride length (**D**). The delta represents the % change from normal to silent running. Correlation coefficients (r) and significance levels (*p*) are described in the separate panels.

**Table 1 sensors-23-02928-t001:** Coefficient of variation (%) from variables extracted during normal and silent running.

Variables	Normal	Silent	*p*
Peak tibial acceleration	14.2 ± 3.71	17.64 ± 3.45 *	**<0.0001**
Heart rate	7.04 ± 3.53	5.60 ± 2.12 *	**0.02**
Cadence	2.38 ± 1.60	3.09 ± 2.70	0.24
Running speed	3.81 ± 1.23	4.75 ± 2.43 *	**<0.05**
Trunk vertical oscillation	5.23 ± 3.11	5.89 ± 3.26	0.38
Stride length	6.79 ± 3.58	6.98 ± 2.60	0.81
Foot contact time	3.98 ± 4.03	5.68 ± 4.97	0.18

* Denotes a significant difference in relation to normal running (*p* < 0.05).

## Data Availability

Raw data are available from the authors upon request.

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
