# Peer review of "Sensitiveness of Variables Extracted from a Fitness Smartwatch to Detect Changes in Vertical Impact Loading during Outdoors Running"

_sensors, 2023, doi:10.3390/s23062928_

Round 1

Reviewer 1 Report

This study assessed whether the data from a fitness smartwatch and chest strap (S&C) is sensible to detect changes in running style. Twenty-eight participants performed 95 m running bouts at 3 m/s in two conditions: normal run-15 ning and running while actively reducing impact sounds.

I suggest to the authors the following:

The tiltes of figures are too long with 4 or 5 lines each

If you want to explain them please explain inside the texts

I suggest to put results in numbers in colculsion

More contect in Mothed section

Author Response

answers uploaded in PDF file

Reviewer 2 Report

Dear Authors

The paper is well written. The work is excellent. 

The question of the utility and trustworthiness of data from smart watches is important. Many people use these without knowing whether the results can be trusted.

The experimental setup is appropriate. The only addition that would have been good would be to have the smartwatch attached to the tibia or shoe.  That would clearly show that it is the location and not the instrumentation that is the problem.  The use of the ‘gold standard’ of the tibial accelerometer is good proof, however.

The statistical analysis is appropriate. Checking for normality is appreciated as too many studies ignore this.

It would be good to have a short discussion of what and how the smart watches should be used and trusted to give useful and accurate results. It might be just where they are worn.

Author Response

answers uploaded in PDF file

Reviewer 3 Report

This manuscript uses a commercial smartwatch paired with its chest strap and tibial-worn accelerometer to investigate whether changes in running technique could be detected by variables extracted from the combination of the smartwatch and chest strap, and whether there was an association between tibial acceleration and the variables from the smartwatch and chest strap(S&C).The results showed a lack of association between biomechanical parameters extracted from the smartwatch and the chest strap and peak tibial vertical acceleration, and that data from the smartwatch combined with chest strap accelerometry were insufficient to monitor changes in running technique.However, the following issues should be addressed prior to publication.

Detailed Comments:

1. The 28 testers selected were overwhelmingly male. Will the gender factor affect the final results of the experiment?

2. What is the reason for setting to 2*95m for short time movement?

3. The paper only used Pearson correlations for calculating the correlation between different data, could more methods be used to mine the correlation between data.

4. Whether the incremental vertical tibial acceleration is correlated with heart rate.

Author Response

answers uploaded in PDF file

Round 2

Reviewer 1 Report

All done for me

Author Response

The authors thank again the reviewer for the relevant work on our submission.